# Transcriptome Characterization and Expression Profiles of Disease Defense-Related Genes of Table Grapes in Response to *Pichia anomala* Induced with Chitosan

**DOI:** 10.3390/foods10071451

**Published:** 2021-06-22

**Authors:** Wanying Hu, Esa Abiso Godana, Meiqiu Xu, Qiya Yang, Solairaj Dhanasekaran, Hongyin Zhang

**Affiliations:** School of Food and Biological Engineering, Jiangsu University, Zhenjiang 212013, China; 2211818004@stmail.ujs.edu.cn (W.H.); 5103181320@stmail.ujs.edu.cn (E.A.G.); 2111818017@stmail.ujs.edu.cn (M.X.); 1000004808@ujs.edu.cn (Q.Y.); 5501800025@stmail.ujs.edu.cn (S.D.)

**Keywords:** biological control, disease resistance, *Pichia anomala*, RNA sequencing, gene ontology, Kyoto Encyclopedia of Genes and Genomes (KEGG)

## Abstract

Transcriptome analysis (TA) was conducted to characterize the transcriptome changes in postharvest disease-related genes of table grapes following treatment with *Pichia anomala* induced with chitosan (1% *w*/*v*). In the current study, the difference in the gene expression of table grapes after treatment with *P. anomala* induced with chitosan and that of a control group was compared 72 h post-inoculation. The study revealed that postharvest treatment of table grapes with *P. anomala* induced with chitosan could up-regulate genes that have a pivotal role in the fruit’s disease defense. The Gene Ontology (GO) and Kyoto Encyclopedia of Genes and Genomes (KEGG) results also confirmed that GO terms and the KEGG pathways, which have pivotal roles in plant disease resistance, were significantly enriched. The up-regulated genes of the treatment group have a unique function in the fruit’s disease resistance compared to the control group. Generally, most genes in the plant–pathogen interaction pathway; the plant Mitogen-activated protein kinase (MAPK) signaling pathway; the plant hormone signal transduction pathway; the pathway of glutathione metabolism; the pathway of phenylalanine, tyrosine, and tryptophan biosynthesis; and the pathway of flavonoid biosynthesis were all up-regulated. These up-regulations help the fruit to synthesize disease-resistant substances, regulate the reactive oxygen species (ROS), enhance the fruit cell wall, and enrich hormone signal transduction during the pathogen’s attack. This study is useful to overcome the lags in applying transcriptomics technology in postharvest pathology, and will provide insight towards developing other alternative methods to using bio-pesticides to control postharvest diseases of perishables.

## 1. Introduction

Transcriptome technologies help to study an organism’s transcriptome through the sum of all its RNA transcripts. The techniques first began in the early 1990s and since have become among the most important techniques in several disciplines [1]. The DNA of an organism’s genome is where the information is contained and expressed through transcription. With the completion of more and more plant and pathogen genome sequencing, and the development of related bio-informatics tools, researchers now have a deeper understanding of the molecular mechanisms of plant–pathogen interaction. RNA sequencing (RNA-seq) is one of the latest techniques using the high-throughput sequencing method to identify genes that are differentially expressed between two samples and has become a powerful method for studying the transcriptome [2]. This technique has been performed for many plant–pathogen interactions, including *Colletotrichum* fungi, *Arabidopsis thaliana* and maize [3], *Magnaporthe oryzae* and rice [4], *Zymoseptoria tritici* and wheat [5], and *Phytophthora nicotianae* and tobacco [6]. However, the application of transcriptomics technology in postharvest pathology lags behind other related fields [7]. It is worth noting that apples [8], sweet oranges [9], pears [10], strawberries [11], etc., have seen their whole genomes sequenced.

So far, little information is available on the molecular and physiological mechanisms by which biological control agents (BCAs) control postharvest diseases in table grapes. However, microbiological, microscopic, biochemical, and molecular techniques have been applied towards the study of other bio-control systems in order to better understand how BCAs function in bio-control [12]. In our previous studies, the microbial and antifungal characteristics of *P. anomala* induced with chitosan against blue mold disease of table grapes caused by *Penicillium expansum* were studied through physical, biochemical, and microscopic techniques. It was also confirmed that chitosan (1% *w*/*v*) is the best inducer of *P. anomala* to control blue mold diseases of table grapes [13]. However, transcriptome techniques can offer a deeper understanding of how the antagonistic yeast can control or inhibit the pathogenic fungi and other major postharvest diseases of the fruit. Therefore, the transcriptome study was very crucial to profile changes in genes related to disease defense.

With this in mind, the study was conducted to determine the transcriptome analysis of table grapes in response to postharvest treatment of *P. anomala* induced with 1% chitosan. This paper also presents the RNA-seq technology and physiological test analysis of postharvest treatment to the response of *P. anomala* in table grapes. Identification of the responsible genes in response to postharvest table grapes treatment with *P. anomala* induced with chitosan can help to understand the possible control mechanisms. In addition, the study is a basis for future research to investigate the postharvest disease control mechanisms for other antagonistic microorganisms.

## 2. Materials and Methods

### 2.1. Grapes

Table grapes (*Vitis vinifera* cv. Red Globe) were obtained from a vineyard in Zhenjiang, Jiangsu Province, China. The fruits were chosen and prepared according to our previous research method [13].

### 2.2. Yeast

*P. anomala* (strain TL0903) was used for all the experiment, and fresh suspension was prepared according to our previous research method [13]. Chitosan (90% deacetylation) was bought from Sangon Biotech Co., Ltd. (Shanghai, China).

### 2.3. Transcriptome Analysis of Table Grapes Treated with P. anomala Induced with Chitosan

#### 2.3.1. RNA Extraction and RNA-Seq Analysis

Uniform wounds (3 mm deep × 3 mm diameter) were made at the midpoint of each grape berry. After that, (1) 15 μL double distilled water and (2) 15 μL *P. anomala* suspension (10^8^ cell/mL) induced with chitosan (1% *w*/*v*) were pipetted into each wound. The grapes were then wrapped with plastic film to keep a high RH (95%) and incubated at 20 °C for 3 days. Samples were taken 72 h post-inoculation. Samples taken from grapes treated with *P. anomala* induced with chitosan were labeled as CY1, CY2, and CY3, and samples collected from the tissues of table grapes treated with sterile distilled water were labeled as CK1, CK2, and CK3. The excised tissues were immediately stored in liquid nitrogen to extract total RNA. RNA was extracted from 2 g frozen table grape tissue (stored at −80 °C) according to the procedure of the Spin Column Plant Total RNA Purification Kit (Sangon Biotech, Shanghai, China). Both the purity and quantity were checked using a spectrophotometer (Thermo Scientific, Waltham, MA, USA) at wavelengths of 260 and 280 nm, respectively. The quality of RNA was evaluated using an RNA Nano 6000 Assay Kit of the Bioanalyzer 2100 system (Agilent Technologies, Santa Clara, CA, USA). A Trizol reagent kit (Invitrogen, Carlsbad, CA, USA) was used to isolate total RNA according to the manufacturer’s instruction with a little modification. 

#### 2.3.2. cDNA Library Construction and Sequencing

After high-quality total RNA was obtained, the mRNA was enriched using Oligo (dT) beads. The mRNA was reverse transcripted into cDNA and amplified using PCR with Phusion High-Fidelity DNA polymerase, distilled water, and primers. Following that, the PCR products were cleaned (AMPure XP system (Beckman Coulter Lifes Sciences, Pasadina, CA, USA) and library quality was determined on the Agilent Bioanalyzer 2100 system. In the end, the cDNA was sequenced using an Illumina Hiseq platform (Genepioneer Biotechnologies Company, Nanjing, China).

#### 2.3.3. RNA-Seq Analysis

High-quality reads were determined using the Perl program. Non-quality reads, reads which have adapters and contain poly-N, were all removed. Then, the high-quality and clean reads were mapped to the corresponding grape genome reference (http://www.genoscope.cns.fr/externe/Download/Projets/Projet_ML/data/12X/assembly/goldenpath/unmasked/, accessed on 24 September 2020) [14]. The expression level was calculated for each gene by Reads Per Kilobase exon Model per Million mapped reads (RPKM), which is currently the most popular and reliable means for estimating gene expression levels. The transcription changes between the control and the treated (*P. anomala* induced with chitosan) samples were determined and identified as DEGs. The treated group’s DEGs that showed higher values than the control were denoted as ‘up-regulated’, while the genes that showed lower values were denoted as ‘down-regulated’.

#### 2.3.4. GO Enrichment Analysis

Gene Ontology (GO) enrichment analysis of the DEGs was conducted by the clusterProfiler R package(R, Boston, MA, USA). Enrichment analysis uses hypergeometric testing to find GO entries that were significantly enriched compared to the entire genome background. Gene Set Enrichment Analysis (GSEA) was also analyzed by clusterProfiler. 

#### 2.3.5. Kyoto Encyclopedia of Genes and Genomes (KEGG) Pathway Enrichment Analysis

KEGG is a database that helps understand high-level functions and utilities of the biological system, such as cells or organisms which have molecular-level information [15]. For the current experiment, KOBAS (2.0) software was used to check the enrichment level of DEGs in the KEGG pathways [16]. The clusterProfiler R packages were used to find KEGG pathways that are significantly enriched compared to the entire genome background.

#### 2.3.6. Validation of RNA Seq Data Using Quantitative Real-Time PCR (RT-qPCR)

A total of 15 DEGs were randomly selected from the RNA-seq data to conduct the RT-qPCR analysis for further confirmation. RNA was extracted from 2 g frozen table grape tissue (stored at −80 °C) using a Spin Column Plant Total RNA Purification Kit (Sangon Biotech, Shanghai, China). Both the purity and quantity were checked using a spectrophotometer (Thermo Scientific, Waltham, MA, USA) at wavelengths of 260 and 280 nm, respectively. The quality of RNA was evaluated using the RNA Nano 6000 Assay Kit of the Bioanalyzer 2100 system (Agilent Technologies, Santa Clara, CA, USA). The first strand of cDNA was synthesized from the RNA using a PrimeScript RT reagent kit with a gDNA Eraser (Takara Biotechnology, Dalian, China) in a PCR System. Specific primers were obtained from Sangon Biotech (Shanghai, China) and are listed in Table 1. The RT-qPCR was conducted with a Bio-Rad CFX96 Real-Time PCR System (Applied Biosystems, Irvine, CA, USA) and the computer program was set according to Wang et al. [17]. RT-qPCR was carried out according to the method describe by Xu et al. [9]. The experiment was conducted twice, and there were three replications per treatment.

## 3. Results

### 3.1. Transcriptome Analysis of Table Grapes Treated with P. anomala Induced with Chitosan

RNA-seq analysis was performed to analyze the gene expression alteration of table grapes in response to postharvest treatment of *P. anomala* induced with chitosan (1% *w*/*v*). Each sample obtained a total of 34,883 reads, which has a GC percentage of approximately 49.82%. Around 96.33% of reads reached the Q20 quality score, and about 82.47% of them were mapped from the grape reference genome.

### 3.2. Functional Enrichment of DEGs

The clustered heatmap clearly shows the frequency in molecular profiling of the DEGs for both samples (control vs. treatment). The red color indicates the number of up-regulated genes, while the blue color shows the number of down-regulated genes. As shown in Figure 1, the DEGs of the tissue of table grapes treated with *P. anomala* induced with chitosan were mostly up-regulated (the red color at the bottom-right portion), and those of the control were mainly down-regulated (the blue color at the bottom-left portion). These results confirm that the postharvest treatment of table grapes with the antagonistic yeast induced with chitosan significantly up-regulated the genetic responses. 

The DEGs of grapes between the control and the group treated with *P. anomala* induced with chitosan (1% *w*/*v*) were determined. A total of 3821 DEGs were screened. Out of them, 2454 genes were up-regulated and 1367 genes were down-regulated (log2 (Fold Change)| ≥1 and FDR < 0.05). A total of 1211 and 226 DEGs were significantly up-regulated and down-regulated, respectively (Figure 2).

The correlation between the two samples was determined using Pearson’s correlation coefficient (r). According to this principle, the closer the correlation value of two treatments to one, the higher the correlation and vice versa. As shown in Figure 3, for the current experiment, there is a high correlation between the two samples. Almost all correlations showed a value of more than 0.9, and only two values were lower than 0.9, with the lowest being 0.721 and the highest being 0.9905.

### 3.3. GO Enrichment Analysis

All the DEGs were analyzed using the GO data, and their function was categorized into three groups: molecular function, cellular component, and biological process. The cellular component group contained 13 GO terms, the molecular function group contained 12 GO terms and the biological process group contained 23 different GO terms. In the cellular component category, the highest classification of differentially expressed unigenes included cell (enriched by 1312 DEGs), cell part (enriched by 1311 DEGs), organelle (enriched by 785 DEGs), membrane (enriched by 670 DEGs), membrane part (enriched by 539 DEGs), organelle part (enriched by 260 DEGs), and extracellular region (enriched by 184 DEGs), which were found to be specific to plant disease-resistance (Figure 4A). 

Of all the GO terms, 12 showed significant enrichment in the molecular function category. Catalytic activity (enriched by 1104 DEGs), binding (enriched by 974 DEGs), transcription regular activity (enriched by 155 DEGs), and transporter activity (enriched by 146 DEGs) were the highest categories of differentially expressed unigenes. These activities have been found to have an important role in plant hormone signal transduction. A total of 23 GO terms were enriched in the biological process category, and these GO terms are the most important to study plant–pathogen defense mechanisms. Generally, metabolic process (enriched by 146 DEGs), cellular response (enriched by 146 DEGs), response to stimulus biological regulation (enriched by 146 DEGs), localization (enriched by 146 DEGs), developmental process (enriched by 146 DEGs), and signaling (enriched by 146 DEGs) subclasses were related to the disease resistance of plants.

To further analyze the possible roles of pivotal DEGs in the induced resistance of table grapes by *P. anomala* induced with chitosan (1% *w*/*v*), the GO subclasses were further subdivided into a tertiary classification (Figure 4B). The major activities which have a significant role in disease resistance and with a significant number of DEGs under biological regulation were regulation of hormone metabolic processes, protein kinase inhibitor activity, positive regulation of cell division, regulation of cell size, signal transduction, regulation of innate immune response, negative regulation of defense response, and activation of protein kinase activity. Response to wounding, response to chitin, response to oxidative stress, plant-type hypersensitive response, and response to fungus are the major activities significantly enriched under the response to stimulus classification. Peroxidase activity and catalase activity were significantly enriched under antioxidant classifications.

### 3.4. The KEGG Enrichment Analysis

To explore the involved biological pathways and functional networks, KEGG enrichment analysis was conducted. Among all the DEGs, 564 DEGs were grouped into 107 KEGG pathways. These pathways were classified into five categories: cellular processes, genetic information processing, environmental information processing, metabolism, and organismal systems (Figure 5). Then, they were further divided into 18 subclasses. The most enriched KEGG pathways of the grape’s transcriptome analysis were biosynthesis of other secondary metabolites (21.10%), carbohydrate metabolism (17%), amino acid metabolism (14%), and environmental adaptation (13.5%) (Figure 5A). Figure 5B shows the top 20 KEGG pathways, which were highly enriched and important in plant–pathogen interaction. Among them, the MAPK signaling pathway of the plant, plant–pathogen interaction, biosynthesis of secondary metabolites, and glutathione (GSH) metabolism have pivotal roles in the disease resistance of plants [18].

To fully understand the biological processes associated with a grape’s reaction to chitosan-induced *P. anomala* treatment, six KEGG pathways were selected. The changes (up-regulation or down-regulation) in genes that encode different enzymes, hormones, and other substances which have significant roles in plant disease resistance/interaction were determined. The selected pathways are the plant–pathogen interaction pathway; the plant hormone signal transduction pathway; the MAPK signaling pathway; the pathway of glutathione metabolism; the pathway of phenylalanine, tyrosine, and tryptophan biosynthesis; and the pathway of flavonoid biosynthesis. The changes in DEGs and their metabolic effect are shown in Figure 6. 

### 3.5. RNA Sequencing Validation by the RT-qPCR

The RT-qPCR experiment was conducted to validate the results obtained from the RNA seq. A total of 15 DEGs were randomly selected for the experiment. The results of the study confirmed that the RT-qPCR results strongly correlated with RNA-seq data (R^2^ = 0.8045) (Figure 7).

## 4. Discussion

RNA-seq (RNA sequencing) is a modern sequencing technique that uses next-generation sequencing (NGS) to discover the presence and amount of RNA in a biological sample at a specific time. In addition, it analyzes the continuous changes in the cellular transcriptome of the sample [1,19]. RNA-seq is widely used in animal experiments; however, plant-based experiments are also widely using these days. For example, RNA-seq analysis has been widely used in *P. expansum*—apple; *Penicillium digitatum*—citrus; and *P. expansum*—pear to understand the transcriptome changes of the fruit during pathogen infection [20,21,22]. However, to the authors’ knowledge, no studies have been conducted on the molecular response of table grapes to microbial antagonists such as *P. anomala*. Therefore, in the current study, the molecular response of table grapes to the antagonistic yeast *P. anomala* induced with chitosan (1% *w*/*v*) was determined.

The RNA-seq result showed that from the total gene reads, 3821 genes were differentially expressed (either up-regulated or down-regulated) (Figure 2). Then, the DEGs were analyzed to determine their GO enrichment. As shown in Figure 4A, the GO enrichment was highly concentrated in the metabolic process, the cellular process, response to stimuli, biological regulation, and signaling. Then, KEGG pathway enrichment analysis was conducted to determine the DEGs which have functional networks and biological pathways in the system. Six KEGG pathways, which have a crucial role in plant–pathogen interaction, are discussed here. However, in the experiment, 107 KEGG pathways were determined for different metabolic activities. 

In plant–pathogen interactions, plants use different mechanisms to protect themselves from pathogen attack. Of all the responses, the earliest one is the response to the generation of an oxidative burst that can induce hypersensitive cell death [23]. This mechanism is called the hypersensitive response (HR) and is among the major elements of plants to control the spread of disease to the other parts. In the current study, enzymes which have a pivotal function in HR, such as CDPK (calcium-dependent protein kinase) and Rboh (respiratory burst oxidase), were enriched after the table grapes were treated with *P. anomala* induced with chitosan. On the other hand, a rapid influx of calcium into the cytosol represents a basis of plant immune responses [24]. The interaction between pathogen perception and calcium signaling is very important for plant pathogen resistance. In the current study, the genes responsible for rapid influx of calcium into cytosol CNGC (cyclic nucleotide-gated channel) were up-regulated because of the postharvest treatment of the fruit with *P. anomala* induced with chitosan. In addition, DEGs which encode CaML (calmodulin) were up-regulated, which plays a significant role in cell wall enlargement and stomatal closure during biotic or abiotic stress (Figure 6A).

As shown in Figure 6B, many metabolic pathways related to the plant MAPK signaling pathway have also been significantly enriched. Some of them, including camalexin synthesis, late defense response to a pathogen, early defense response to a pathogen, and maintenance of the homeostasis of ROS, which has a direct relation with plant–pathogen interaction, were all enriched. The DEGs encode a BAK1 (brassinosteroid insensitive 1-associated receptor kinase 1) enzyme, which plays a pivotal role in the metabolic pathway of plant cell death defense response, and camalexin synthesis was up-regulated [25]. Other substances and enzymes which were enriched during the postharvest table grapes’ treatment with *P. anomala* enriched with chitosan include WRKYY3 (WRKY transcription factor 33), WRKYY2 (WRKY transcription factor 22), MPK3 (mitogen-activated protein kinase 3), PR1 (pathogenesis-related protein 1), ACS6 (1-aminocyclopropane-1-carboxylate synthase), CaM4 (calmodulin), and RbohD (respiratory burst oxidase). For example, the enrichment of RbohD makes the fruit immediately release ROS (such as superoxide anion and hydrogen peroxide) from different cells, which is one of the plant defense mechanisms during pathogen attack. 

Different genes were up-regulated in the plant hormone signal transduction pathway. In this pathway, the DEGs which encoded NPR1 (regulatory protein NPR1), TGA (transcription factor TGA), and PR-1 (pathogenesis-related protein 1) were all up-regulated (Figure 6C). PR-1 is a protein produced during pathogen attack and used as systemic resistance against the pathogen. It has antimicrobial characteristics and can attack the cell wall of bacteria or fungi [26]. NPR1 is known for its key regulation of the activities of the salicylic acid (SA)-mediated systemic acquired resistance (SAR) pathway. It helps the plant tissue to resist pathogen attacks [18]. TGA has different biological processes, such as the regulation of growth and development, resistance against pathogen attacks, and abiotic stress [25]. Other genes that encode enzymes, and other substances which have a crucial role in the fruit biological process, were also up-regulated. Some of them, including TCH4 (xyloglucan: xyloglucosyl transferase TCH4, which has a crucial role in cell elongation), CYCD3 (cyclin D3), AHP (histidine-containing phosphotransfer protein, which plays a crucial role in cell division), GH3 (auxin-responsive GH3 gene family), and SUHR (SAUR family protein), are detrimental for cell enlargement and plant growth (Figure 6C).

Generally, increment of glutathione in peroxisomes and chloroplast can induce disease resistance and tolerance at the early stage of plant–pathogen interaction [27]. As shown in Figure 6D, postharvest treatment of table grapes with *P. anomala* induced with chitosan stimulated the enzymes and other substances involved in the pathway of glutathione metabolism and glutathione accumulation. It helps as an antioxidant in plants and can prevent the damage to cellular components caused by the ROS. GST (glutathione S-transferase), GGT1_5, CD224 (gamma-glutamyltranspeptidase/glutathione hydrolase/leukotriene-C4 hydrolase), speE, SRM, SPE3 (spermidine synthase), ODC1, speC, and speF (ornithine decarboxylase) are among the major enzymes up-regulated during the postharvest treatment of table grapes and have a pivotal role in glutathione metabolism.

As shown in Figure 6E, postharvest treatment of table grapes with *P. anomala* induced with chitosan stimulated the enzymes and other substances involved in the pathway of phenylalanine, tyrosine, and tryptophan biosynthesis. ADT, PDT (arogenate/prephenate dehydratase), PAT, ATT (bifunctional aspartate aminotransferase and glutamate/aspartate-prephenate aminotransferase), GOT1 (aspartate aminotransferase, cytoplasmic), and TAT (tyrosine aminotransferase) are the DEGs that encoded the enzyme and other substances involved in the pathway which were all up-regulated. These up-regulations enrich the activities and concentration of prephenate, l-arogenate, phenylalanine, phenyl-pyruvate, etc., which have a critical role in natural plant disease defense. For example, phenylalanine accumulation induces the enzyme phenylalanine ammonia-lyase (PAL), which has a crucial role in the Shikimate acid pathway leading to the biosynthesis of phenols, phytoalexin, and lignins. All these secondary metabolites are important in plant disease resistance. 

Flavonoids and other phenolics have a pivotal role in plant disease resistance during pathogen infection, which act as phytochemicals in plants and inhibit the germination of pathogen spores when plants are infected by pathogens [9]. In the flavonoid biosynthesis pathway of table grapes, the expression levels of CHS (chalcone synthase), CYP73A (trans-cinnamate 4-monooxygenase), PGT1 (phlorizin synthase), E2.3.1.133, and HCT (shikimate O-hydroxycinnamoyltransferase) were enriched by P. anomala induced with chitosan (Figure 6F). For example, the enzyme CHS, enriched in multiple paths in the flavonoid biosynthesis pathway, is a key enzyme of the flavonoid/isoflavonoid biosynthesis pathway [28]. It has a paramount function in the plant natural disease defense mechanism, and also has uses as synthetic intermediates. CHS expression results in the accumulation of flavonoid and isoflavonoid phytoalexin, and it has a significant role in the salicylic acid defense pathway.

## 5. Conclusions

In conclusion, postharvest treatment of table grapes with *P. anomala* induced with 1% chitosan can up-regulate genes which have pivotal roles in fruit disease defense. Generally, most genes in the plant MAPK signaling pathway; plant–pathogen interaction pathway; plant hormone signal transduction pathway; the pathway of glutathione metabolism; the pathway of phenylalanine, tyrosine, and tryptophan biosynthesis; and the pathway of flavonoid biosynthesis were all up-regulated. These up-regulations help the fruit to synthesize disease-resistant substances, regulate the ROS and enhance the fruit cell wall, and enrich hormone signal transduction during pathogen attack or other biotic stresses.

## Figures and Tables

**Figure 1 foods-10-01451-f001:**
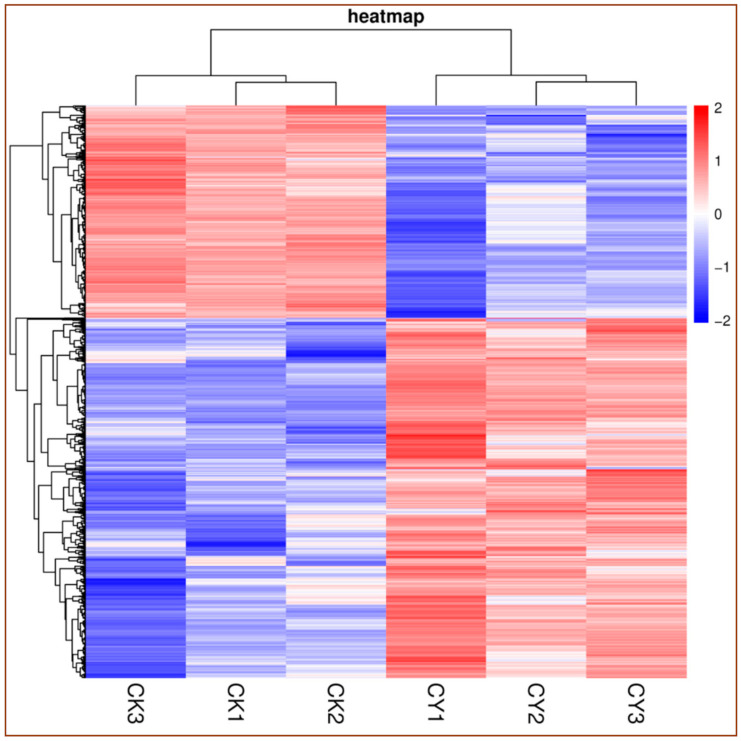
Heat map of DEGs for control vs. table grapes treated with *P. anomala* induced with chitosan (1% *w*/*v*). CK = control, CY = *P. anomala* induced with chitosan. The red color shows the number of up-regulated DEGs, and the blue one indicates the number of down-regulated DEGs.

**Figure 2 foods-10-01451-f002:**
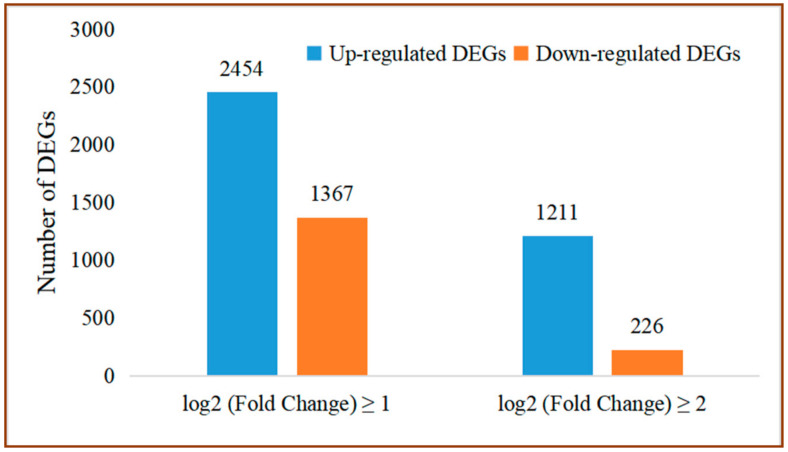
Number of DEGs between treated and control samples.

**Figure 3 foods-10-01451-f003:**
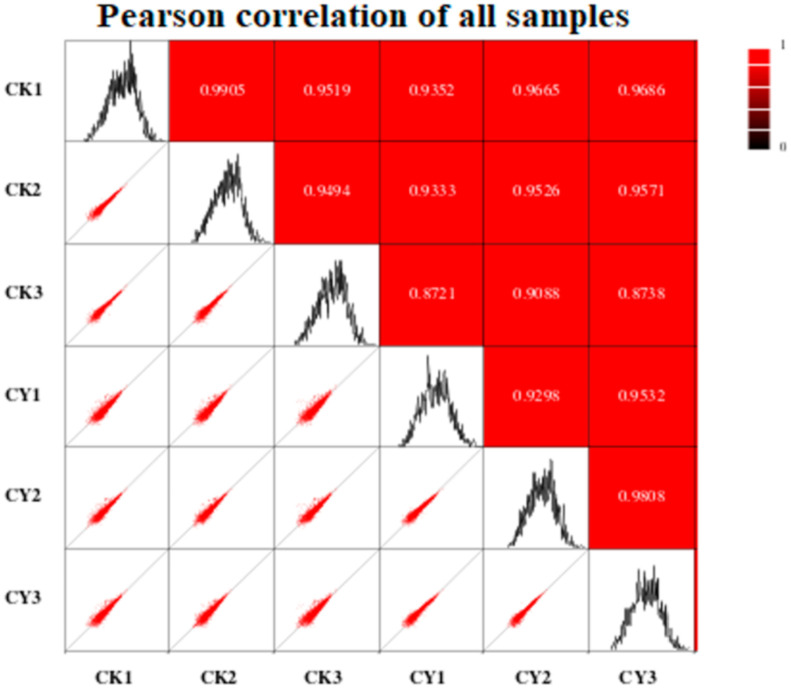
Pearson’s correlation coefficient between treated and control samples; the higher the correlation between the two samples, the closer the value is to 1. CK = control and CY = *P. anomala* induced with chitosan (1% *w*/*v*).

**Figure 4 foods-10-01451-f004:**
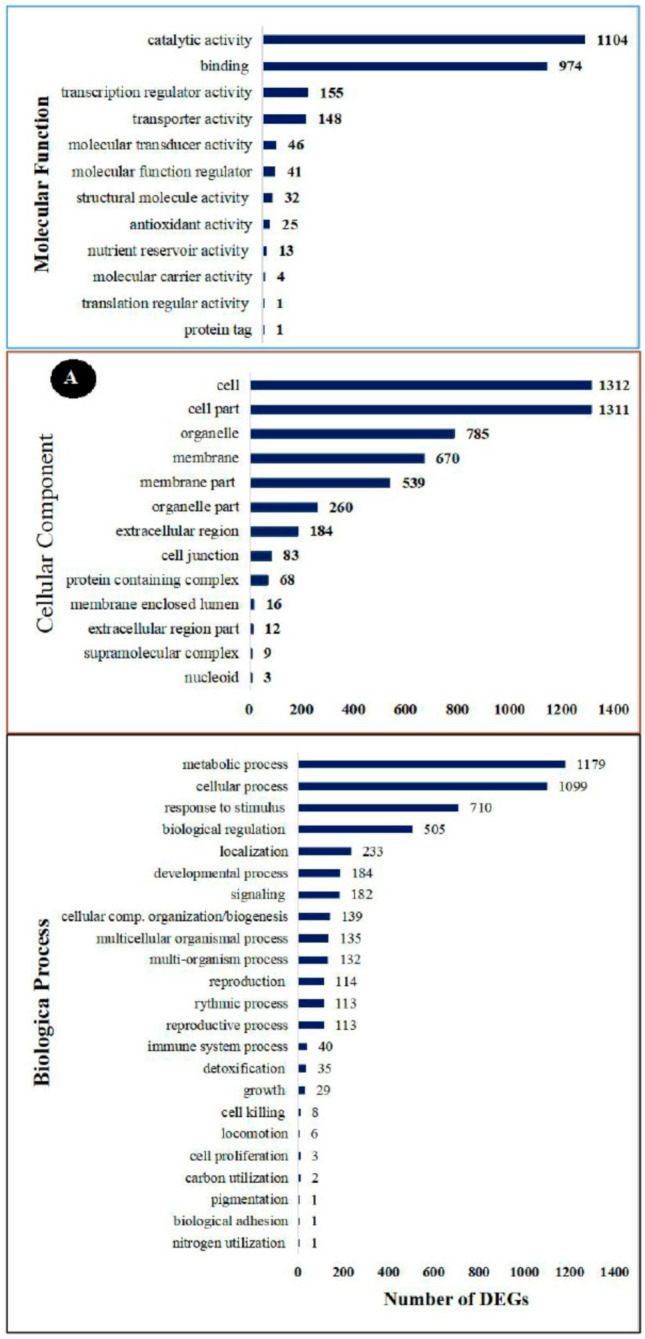
The GO enrichment analysis of significantly enriched DEGs in the RNA-seq result. (**A**) Secondary classification. (**B**) Tertiary classification.

**Figure 5 foods-10-01451-f005:**
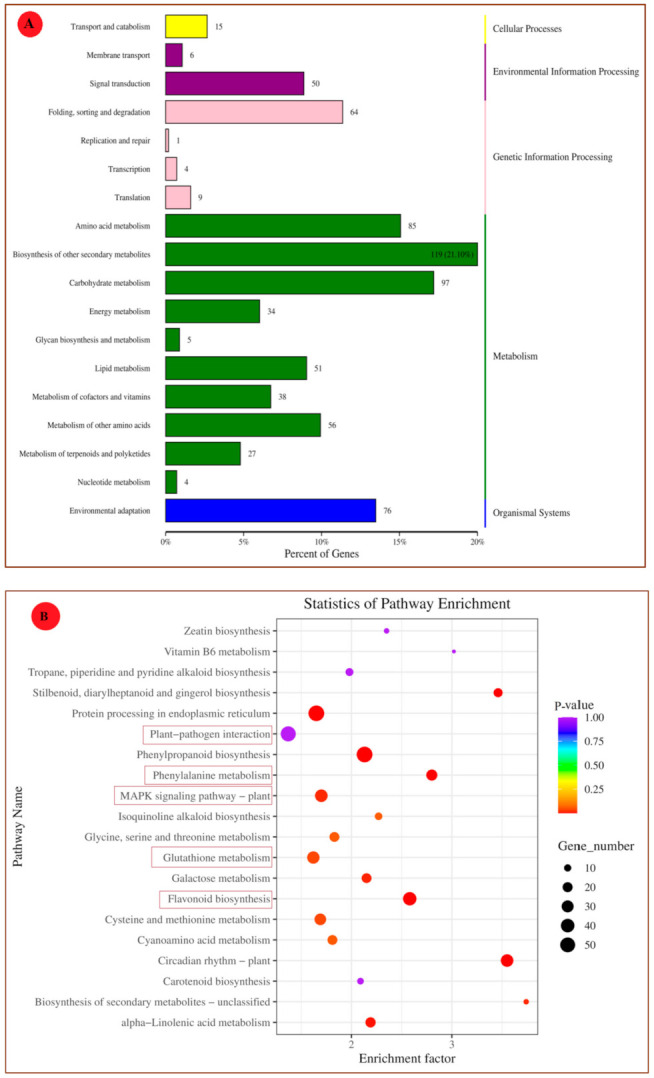
The KEGG pathway enrichment analysis (**A**) Secondary classification. (**B**) Top 20 enriched pathways for table grapes treated with *P. anomala* induced with 1% chitosan after 72 h.

**Figure 6 foods-10-01451-f006:**
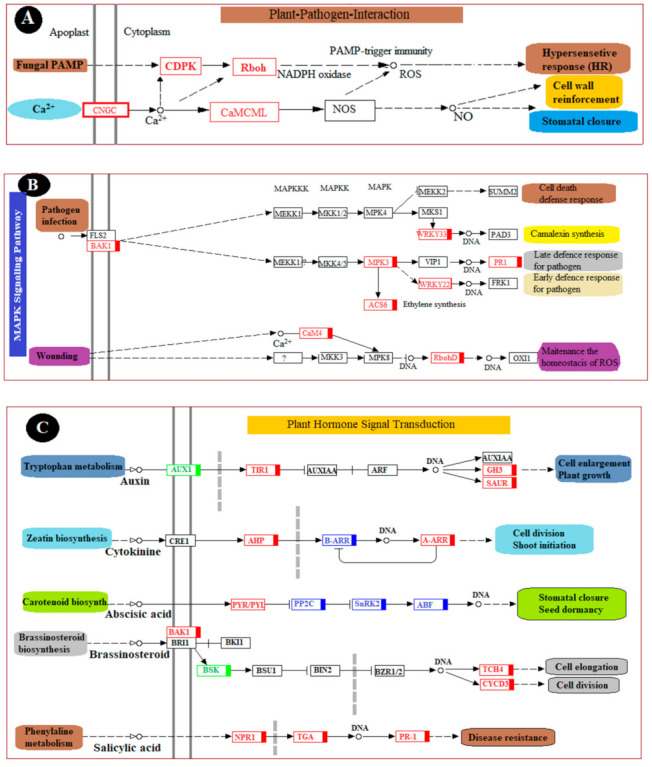
Major pathways related to induced disease resistance of table grapes. (**A**): Plant–pathogen interaction pathway, (**B**): Plant MAPK signaling pathway, (**C**): Plant hormone signal transduction pathway, (**D**): Pathway of glutathione metabolism, (**E**): Pathway of phenylalanine, tyrosine, and tryptophan biosynthesis, and (**F**): Pathway of flavonoid biosynthesis. The genes with the red color are all up-regulated, the genes with the green color are all down-regulated, and the genes with the blue color are mix-regulated.

**Figure 7 foods-10-01451-f007:**
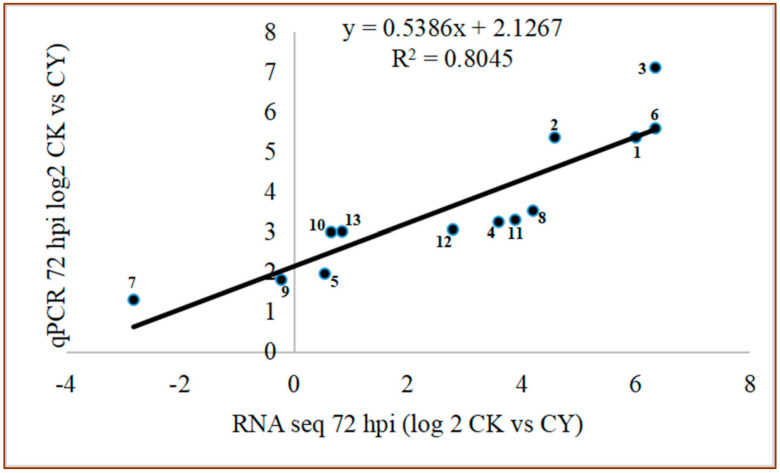
The comparison of relative gene expression level between RNA seq and RTq−PCR. CK = Control, CY = *P. anomala* induced with chitosan (1% *w*/*v*), 72 hpi = 72 h post inoculate.

**Table 1 foods-10-01451-t001:** Primers used for the RT-qPCR to confirm the transcription result of random genes obtained by RNA-seq.

Gene	Primer Sequence (5′→3′)
VIT_203s0063g01160-F	GCTCCGTCCGCTTCAATCACTAC
VIT_203s0063g01160-R	TCCAACCCTCCGACATCATCCTG
VIT_203s0063g01150-F	AGAGGTCGATCCAGCCTTCATCC
VIT_203s0063g01150-R	TGGTTGGAAGCATTGGCGGAAG
VIT_208s0056g00780-F	GCAGCAGCAGATCAAGGAGAAGG
VIT_208s0056g00780-R	CCGATGAACGCCTGGCTGATG
MSTRG.2525-F	TCCCTCCCTCTACGCCCTCTC
MSTRG.2525-R	AACTCCGCTTGCTCCCTCTACC
VIT_214s0060g02170-F	GGGCAAGGTTCTGGGCTAAGTTC
VIT_214s0060g02170-R	GCTTCCCTTTGCTCCTTCTCTTCC
MSTRG.16261-F	TCAGCAGTGAGTAAGAGGCAGAGG
MSTRG.16261-R	GTTCCGCCGCCTAACCTTGAAG
MSTRG.10321-F	ACCTGGGCTCGGGCAACTTC
MSTRG.10321-R	GATTAGCGGCGGCAGTGACG
MSTRG.7282-F	TGCCAAGTGGGTCAAAAGCCTTAG
MSTRG.7282-R	ATTGCCGGAAACTTACGTGAGGTG
MSTRG.3971-F	TCCTTCCAATCAATGCGAGTGCTC
MSTRG.3971-R	CAACACGTTCGACCTGTGGGATC
VIT_200s1458g00010-F	GGAGGGAGGGGAAAGCAACTAATG
VIT_200s1458g00010-R	TTGCACGCATAGGAGCCCAATG
VIT_217s0000g08030-F	TGACTTCAAGCACCCGATAAACCC
VIT_217s0000g08030-R	GCAGCAGTATGGACCTCTGAGTTG
VIT_210s0116g00080-F	ATCCTTCTGGGCTGCCTACTCTG
VIT_210s0116g00080-R	TCTACTGCTGGAATCCGCCTACC
VIT_218s0001g00850-F	GCATCGGTTCAGGTCATGGCTAG
VIT_218s0001g00850-R	GTTTAGTGCTCGGGCTCAATGGG
VIT_213s0019g03300-F	CGGCAACGACATTCTGGATACCTC
VIT_213s0019g03300-R	ATCCCAAGATCCCGTAGTCCCAAG
MSTRG.3509-F	GATTGGGCACGGGAATGGTCAG
MSTRG.3509-R	AGGAAGAATGGTGAACGTGTCTGC
ACTIN-F	TTCAATAAGGAGAAGATGGTGGA
ACTIN-R	TTGGTGAGGTAGTCTGTGAGGTC

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
