# Peer review of "Transcriptome Characterization and Expression Profiles of Disease Defense-Related Genes of Table Grapes in Response to Pichia anomala Induced with Chitosan"

_foods, 2021, doi:10.3390/foods10071451_

Round 1
Reviewer 1 Report
The manuscript is well written but initial abstract, introduction and M&M sections needs improvement in the language. The places with possible changes have been highlighted in green.
Few other changes suggested are as below:
Title: Line 3, 4 : Table and induced spellings mentioned with hyphen?
Abstract:
Line 10-12: The sentence would have been more clear as – “The transcriptome analysis (TA) was conducted to characterize the transcriptome changes in postharvest disease related genes of table grapes following the treatment with Pichia anomala induced with chitosan (1% w/v).” There was no need to use indirect speech. I feel it just complicated the statement.
Line 11: Pichia anomala in italics and no need of hyphen in between. Similarly, P. anomala should be in italics throughout the manuscript.
Line 13: over treatment? Do you mean following the treatment?
Line 13: no need to ‘after’ if ‘post-inoculation’. …compared after 72 hours of inoculation.
Line 16: GO, KEGG, MAPK – use full forms when used for first time in manuscript.
Keywords: Try using keywords which are not repeated in the title.
The discussion and conclusion parts are well written.
Author Response
Dear reviewer,
Please check the attached file as responses to your valuable comments and suggestions.
With best regards,

Reviewer 2 Report
In general, the study is connected to the journal's objectives. The study is very interesting and novel and the use of the transcriptome was adequated. However, the manuscript has many text errors that need to be corrected and is necessary to include some information.
In the following, I give a detailed revision of the manuscript.
- All scientific names must be in italics
- Many words have hyphens where they should not, please adjust the whole document
- I suggest that the incubation stage of the microorganism should be separated from the sample treatment and RNA extraction.
- How is RNA quality quantified? (Line 101)
- Was the qPCR efficiency of the primers calculated? and if so, how was it done? Please included the information.
- I consider it necessary to include or cite your previous works where the reason for having used 1% chitosan is explained
- Line 164: Please define the abbreviation
- It is necessary to improve the resolution of figure 4
Best regards
Author Response

(The authors gave the same response as above.)
